# Effects of Water and Wind Stress on Phytochemical Diversity, Cannabinoid Composition, and Arthropod Diversity in Hemp

**DOI:** 10.3390/plants14030474

**Published:** 2025-02-05

**Authors:** Ericka R. Kay, Casey S. Philbin, Lora A. Richards, Matthew L. Forister, Christopher Jeffrey, Lee A. Dyer

**Affiliations:** Department of Biology, University of Nevada Reno, 1664 N Virginia St., Reno, NV 89557, USA; ekay@unr.edu (E.R.K.); cphilbin@unr.edu (C.S.P.); lorar@unr.edu (L.A.R.); mforister@unr.edu (M.L.F.); cjeffrey@unr.edu (C.J.)

**Keywords:** phytochemical diversity, hemp, cannabinoids, CBD, drought stress, flood stress, arthropod diversity

## Abstract

Phytochemical diversity is increasingly appreciated as an important attribute of plants that affects their interactions with other organisms and can have substantial effects on arthropod communities, but this axis of diversity is less studied for agricultural plants. For both managed and natural systems, understanding how extreme weather events, such as droughts, floods, and extreme wind, affect phytochemical diversity is an important part of predicting responses of plant–arthropod interactions to climate change. In an outdoor field experiment with two distinct varieties of hemp (*Cannabis sativa* L., Cannabaceae), we investigated the effects of simulated water stress from reduced water availability and flooding, along with an unplanned extreme wind event on phytochemical diversity and cannabinoid profiles. We also examined how changes in chemistry affected the diversity of the associated arthropods. Our results indicate that both genetic variety and environmental stress have substantial effects on variation in hemp phytochemical diversity and cannabinoid composition, and these effects cascaded to alter the arthropod communities on flowers. The largest differences in chemistry were found between different varieties, which accounted for over 10% of the variation in phytochemical diversity. Stress from wind and floods reduced the phytochemical diversity of flowers, wind had negative effects on cannabidiol (CBD) concentrations, and both water deficit and flooding caused subtle shifts in cannabinoid composition. The subsequent cascading effects of chemistry depended on how it was characterized, with increases in CBD causing higher arthropod richness, while increased phytochemical diversity reduced arthropod diversity. These results provide insights into the potential effects of extreme weather on hemp chemistry, as well as the consequences of hemp phytochemical diversity on colonizing arthropods.

## 1. Introduction

Phytochemical diversity has become a focus of modern chemical ecology, due partly to a greater recognition of the advantages of defensive redundancy (i.e., having multiple secondary metabolites with putatively similar defensive roles) and advances in metabolomics [1,2]. There are numerous adaptive and nonadaptive hypotheses related to the intraspecific redundancy and diversity of plant-specialized metabolites [2,3], but antiherbivore defense is one of the most focused areas of research on redundancy [4]. The higher diversity of phytochemical profiles can deter specific herbivores but can also enhance the diversity of associated arthropods [2,3]. In the context of agricultural crops, phytochemical diversity can be managed via resource inputs to reduce pest damage but also to improve yield of focal compounds, such as caffeine in *Coffea* (Rubiaceae) crops. For cannabis cultivated for medicinal purposes in particular, maintaining consistent cannabinoid and terpene profiles is essential, as even minor changes in gene expression can impact potency, efficacy, and product stability. Drought stress has been shown to increase concentrations of major cannabinoids like tetrahydrocannabinolic acid (THCA) and cannabidiolic acid (CBDA) by 12–13% [5]. Short-term drought stress during early flowering can alter cannabinoid profiles, decreasing cannabidiolic (CBD) and tetrahydrocannabinolic (THC) while increasing cannabigerolic (CBG) by 40%, and heat stress can reduce cannabigerolic acid (CBGA) production in young cannabis flowers [6].

An important goal of chemical ecology and agricultural sciences is to narrow the knowledge gap about how chemical profiles shift under changing resource availability, including water and nutrient resources, and how these shifts affect arthropod communities. Closing this gap will contribute to predicting how plants will respond to future climate scenarios, such as increases in drought and floods. This is particularly critical for hemp producers to remain compliant with legal standards and prevent the loss of crops to pest insects. Here, we examine the effects of water availability on the phytochemical profiles of agricultural *Cannabis sativa* L. (Cannabaceae, hereafter referred to as hemp) and how these changes affect the associated arthropods.

*Cannabis sativa* is a wind-pollinated, short-day flowering, dioecious, annual plant that is cultivated for its fiber, seed oil, and phytochemical production. It is classified into different chemotypes based on its chemical profile, with varieties categorized as high-THC, high-CBD, or intermediate types. To increase flower production and maximize cannabinoid yields, pollination is prevented, and only female plants are grown. Research on hemp has increased substantially in recent decades due to the growing interest in its medicinal and industrial applications [7,8]. The chemistry of hemp cannabinoids has been the focus of a myriad of different studies [9], and cannabinoid biosynthesis and other reaction pathways are relatively well understood; this is true for the relationships between cannabidiol (CBD), cannabidiolic acid (CBDA), cannabidivarin (CBDV), cannabigerolic acid (CBGA), cannabinol (CBN), Δ-8-tetrahydrocannabinol (Δ8-THC), and Δ-9-tetrahydrocannabinol (Δ9-THCA), as well as other known compounds (Figure 1). However, there is much to learn about how cannabinoid pathways can shift under different environmental conditions, and there is a substantial knowledge gap about the biochemistry and chemical ecology of the full spectrum of hemp secondary metabolites [10,11]. In fact, the “entourage effect” [12] is a hypothesized synergy between cannabinoids and terpenes that warrants further applied investigation [13]. Does this “entourage effect” impact arthropods and other interacting organisms?

Untargeted metabolomic approaches are likely to offer new insights into these issues and into the ecological functions of hemp chemistry, as well as potential uses. The Agricultural Act of 2018, also known as the Farm Bill, legalized hemp production federally in the United States, resulting in an expansion of production [14,15,16]. This expansion has led to hemp becoming an increasingly common species across the landscape and a potential resource for generalist pests and other arthropods. As a relatively novel plant in the United States, introduced to North America in the 18th century [7,17], it is another important, interesting, and understudied aspect of global change that can affect biotic communities. For example, the corn earworm (*Helicoverpa zea* Boddie, Noctuidae) is a major hemp pest, and hemp infestations by the larvae of this species can have damaging consequences to crop yields by increasing the levels of CBD and THC beyond the 0.3% legal limit [18].

To address knowledge gaps on the chemical ecology of hemp, we designed an experiment to test how varying water stress levels affect cannabinoid biochemical pathways, phytochemical diversity, and associated arthropod diversity. We hypothesized that water stress, which is an increasingly common attribute of climate change, alters hemp’s chemistry by both altering phytochemical diversity and shifting the specific biochemical pathways involved in cannabinoid synthesis. These changes are expected to have cascading effects on interactions with other organisms, particularly colonizing arthropod communities.

## 2. Results

### 2.1. Phytochemical Diversity

Phytochemical diversity was affected most by variety (C versus L), with C plants characterized by considerably lower diversity under all treatment level concentrations (Figure 2; also see Appendix A), but this genetic effect was smaller in the SEM (Figure 3). The magnitude of effects in the causal models (SEM) was different due to appropriate control of confounders [19]. All SEM coefficients reported here are standardized and are reported as “causal effect sizes” since they are derived from causal models and provide estimates of the expected change in the response variable (in standard deviation units) for each one standard deviation change in the predictor variable, holding other relationships in the model constant. Stress responses caused by water deficit and wind damage were weaker than variety, but both forms of stress caused higher levels of phytochemical diversity (Figure 3). These differences in phytochemical profiles suggest that both genetic variety and environmental conditions (in this case, water availability) interact to shape the phytochemical landscape of hemp. Effect sizes for phytochemical diversity are difficult to interpret, but the shifts seem large, with raw differences in diversity due to variety as high as 150% (Figure 2), but when the appropriate causal structure is included (i.e., statistical controls are correctly applied), the difference is about 55% (Figure 3). Decreases in phytochemical diversity due to wind and water stress were about 5% of a standard deviation (Figure 3). Based on the Bayesian multiple regression, traditional statistical interactions were negligible, with similar patterns of effects of stress across varieties (Figure 2).

### 2.2. Arthropods on Hemp

Diverse communities of arthropods were collected from the flowers of experimental plants (abundances are summarized in Appendix A). The most common orders were Thysanoptera and Hemiptera, with very high abundances of herbivore families, most notably Thripidae, Aphididae and Cicadellidae. Other abundant herbivorous taxa included Chrysomelidae (Coleoptera) and Rhopalidae (Hemiptera). Common predaceous or parasitoid families included Coccinellidae (Coleoptera), Anthocoridae and Pentatomidae (Hemiptera), and Braconidae and Encyrtidae (Hymenoptera). Omnivores included Formicidae (Hymenoptera) and Syrphidae (Diptera—these are predaceous as larvae).

Treatments affected the diversity and frequency of arthropod visitors, with specific patterns visible across different treatment levels and plant varieties. The SEMs provided estimates of causal relationships between arthropod diversity, cannabinoids, flood, drought, wind stress, and variety (Figure 3). Again, all coefficients reported here are standardized and provide estimates of the expected change in the response variable (in standard deviation units) for each standard deviation change in the predictor variable, holding other relationships in the model constant. Effects of exogenous variables on phytochemical diversity and arthropod richness were considerable, with wind stress (−0.71) and flooding (−0.1) reducing plant height considerably (Figure 3). These effects on plant size and those on phytochemical diversity, reported above, cascaded to affect arthropod diversity, which was positively associated with plant height (0.09) and negatively affected by chemical diversity (−0.22). Again, the results suggest that both genetic and environmental factors shape metabolic and arthropod diversity on hemp.

### 2.3. Cannabinoid Diversity

Cannabinoid concentrations (Appendix A) fell within expected ranges for hemp [10] (Fischedick et al. 2010). We estimated cannabinoid diversity, based on quantification of CBD, CBDA, CBDV, CBGA, Δ8-THC, and Δ9-THCA. We analyzed the three commonly used Hill numbers (richness, Shannon, Simpson) diversity and show results from Simpson’s diversity (q = 2) since all compounds were present in all samples (i.e., there are no rare peaks). We also estimated concentrations of a single cannabinoid, CBD, and a latent factor (Factor 1 in Appendix A) representing the combined measured cannabinoids (CBD, CBDA, CBDV, CBGA, Δ8-THC, and Δ9-THCA). (Figure 4 shows results for CBD.) When CBD was modeled as the focal endogenous variable, it was strongly affected by variety (0.24), with L exhibiting higher CBD levels than C (Figure 4). CBD had a positive direct effect on arthropod richness (0.09), suggesting that higher CBD levels may promote greater arthropod diversity. Direct negative effects of flood (−0.16) and wind stress (−0.71) on plant height indirectly affected CBD levels and arthropod richness, illustrating the complex relationships between abiotic stressors, plant architecture, and secondary chemistry in determining arthropod community composition. The same models with cannabinoid diversity, other individual cannabinoids, the latent factor, or a traditional SEM with a latent cannabinoid factor all yielded similar results (Appendix A).

The biochemical pathways SEM revealed subtle shifts in all metabolic pathways of cannabinoids across the different varieties (C, L) (Figure 5) and water treatments (water stress, control, and flood) (Appendix A).

## 3. Methods

### 3.1. Experimental Design and Plant Cultivation

This experiment was conducted at the University of Nevada, Reno’s (UNR) outdoor greenhouse facility in 2020 (Appendix A). One hundred fifty female clones bred for CBD production were planted in 100-gallon fabric pots, commonly used in *Cannabis* cultivation; propagating from clonal cuttings is a common practice in cannabis cultivation to ensure genetic uniformity. In 2019, the pots were initially filled to 80% of capacity with a 1:1 ratio mixture of SOAR Potting Mix and Tahoe Propagation Mix (Full Circle Compost) and mulched with rice straw for a different hemp experiment. These same pots, soil and mulch were left in place and used again in 2020. All hemp plants were USDA-compliant and contained no more than 0.3% THC dry weight. On 27 June, four days before planting was scheduled, a wind event occurred while the plants were outdoors while undergoing hardening off in their smaller pots. Hardening off is a critical step to acclimate plants to outdoor conditions and prevent high mortality after planting. The wind event resulted in the death of 110 clones, including both Cherry Wine and Lifter varieties. Due to the loss of clones, we needed a replacement quickly. Cherry Wine clones were the only variety we could source within a reasonable driving distance, resulting in an unbalanced treatment representation with two hemp varieties: 108 Cherry Wine (C) clones and 42 Lifter (L) clones, which were planted on 1 July 2020. We randomly assigned plants to 3 treatment levels designed to simulate “water deficit” (lower water availability), “control” (normal watering methods), and “flood” (excess water applied) in a fully factorial experimental design (Appendix A). These treatments were maintained consistently throughout the growing season. At planting, all plants were thoroughly hand-watered. Water was supplied via an automated drip irrigation system, with 3 emitters per plant spaced evenly around the center that had flow rates set to 0.5, 1, and 2 gallons per hour for the deficit, control, and flood treatments, respectively. Drip irrigation was applied three times a week using a timer starting on week one. Additionally, to ensure plants would establish, plants were hand-watered 3 times a week for a count of 30 s at full pressure for approximately the first 30 days on the same day as the drip irrigation was applied. Approximately one month post-planting (1 August), it was observed that the 40 plants that survived the wind event did not grow vegetatively and instead prematurely entered the flowering growth stage, likely triggered by wind-related stress. This resulted in a mensurative experimental manipulation: wind stress.

On 11 August 2020, hand-watering ceased, and each treatment group received one hour of water drip irrigation (1.5, 3.0, 6.0 gal) each morning. Prior to initiating water treatments, leaf counts and height measurements were again recorded to capture any initial variation between plants. Once a week, an additional 2.5 gallons of water was applied to each plant to ensure uniform soil wetting and prevent the loss of drip irrigation via channeling. The amount of water needed to adequately moisten the soil was calculated by timing the duration required to fully hydrate a pot. A 30 s application, equivalent to 2.5 gallons, was determined as sufficient. To ensure drip emitters were watering accurately, periodically throughout the season, a subset of emitters was tested by placing a pot’s three emitters in a 5 gal bucket for two minutes and measuring the water released.

Water stress was evaluated using an infrared (IR) thermometer (Extech Instruments Model RH401, Nashua, NH, USA) with an emissivity setting of 0.95. The spot-to-distance ratio was 8:1, and measurements were taken at a 4-inch distance from the fan leaf, providing a field of view of approximately 0.5 inches. Care was taken to avoid detecting energy from the soil, nearby plants, or the sky, and leaves were not shaded by the IR thermometer to prevent changes in surface temperature. The IR reading was recorded once the thermometer reached a stable temperature. Soil moisture was assessed using a moisture meter, with the probe inserted two-thirds of its length into the soil, positioned midway between the pot perimeter and plant stem.

### 3.2. Insect Sampling and Plant Harvest

To assess arthropod interactions with the plants, insects were sampled by hand and by beating sheets from all 150 plants and were immediately frozen. The entire plant was sampled, and beating was standardized. Throughout the growing season, plants were randomly selected for these arthropod surveys, with a final survey conducted on each plant prior to harvest. These arthropods, mostly insects, were later identified to the lowest possible taxonomic level in the laboratory. Phytochemical diversity is likely to affect community composition in interesting ways [2], so arthropod diversity was estimated as the richness per plant to focus more on the community composition of an individual plant and to minimize errors associated with estimating the abundances of individual species. Harvesting was conducted at the end of the growing season; inflorescences were collected from three canopy levels—the lower, middle and upper—with the upper canopy sample taken from the dominant upper cola. All biomass samples were dried in paper bags for subsequent chemical analysis (Appendix A).

Throughout the growing season, plants were randomly selected for arthropod surveys, with a final survey conducted on each plant prior to harvest. Surveys included visual searches followed by sampling by a beat sheet. Arthropods were collected in 3-ounce plastic portion cups or with an aspirator and frozen for later identification. A representative specimen of each morphotype was preserved in a reference collection at the University of Nevada Museum of Natural History. The second week of October, final inflorescence samples were collected and dried in brown paper bags under laboratory conditions to preserve their integrity for chemical analysis.

### 3.3. Chemical Analysis

All extractions, standards preparation, and chromatography were carried out using Optima grade Methanol (Fisher Scientific, Waltham, MA, USA), 18 MΩ water (Thermo Smart2Pure Pro, Waltham, MA, USA), and spectral-grade ammonium acetate (LiChropur, Millipore-Sigma, Burlington, MA, USA). Cannabinoid standards were obtained as 1 mg/mL solutions in MeOH from Agilent (Santa Clara, CA, USA) and pooled to constant concentration (3.00 × 10^−4^ M in methanol) for generating dilution series.

Plants were analyzed for both untargeted metabolomics and targeted cannabinoid quantification (summarized results are provided in Appendix A), adapted from the previously reported extraction protocols [20]. Methanol was chosen to enhance the extraction of more polar compounds without significantly sacrificing cannabinoid extractability. Samples were finely ground using a TissueLyser II (Qiagen, Hilden, Germany) and 10.0–12.0 mg was extracted in 2.00 mL of methanol. Following a 10 min sonication and an overnight wrist-action shake, extracts were centrifuged and filtered through 0.25 μm PTFE syringe filters then diluted 1:10 with internal standard stock (10.0 μM umbelliferone in methanol). Extracts (1.0 μL) were injected on an Agilent Infinity 1260 II UPLC (Agilent, Santa Clara, CA, USA), coupled to a 6546A high-resolution QTOF mass spectrometer with electrospray ionization in negative scan (TOF only) mode, and equipped with an Agilent Poroshell reversed-phase column (Phenyl-Hexyl, 2.1 × 150 mm, 1.9 μm) using a water (solvent A) and methanol (solvent B) gradient, both containing 10.0 mM ammonium acetate. The gradient began with initial conditions of 20% B; ramping up to 60%B (0–6 min), 70% B (6–8 min), 70–85% B (8–18 min), and 85–100% B (18–19 min); holding at 100% B (19–21 min) before reconditioning the column at 20% B (21.1 min–23 min).

Peaks, analyzed in scan mode, with a height greater than 20,000 counts were extracted, and compounds found to have a peak height greater than 100,000 in fewer than two individuals were removed. Peak areas normalized to the internal standard area and plant dry mass were then used to estimate overall phytochemical diversity. Calibration curves were generated in scan mode for cannabinoid standards using the same LC-MS conditions described above. We quantified cannabidiol (CBD), cannabidiolic acid (CBDA), cannabidivarin (CBDV), cannabigerolic acid (CBGA), cannabinol (CBN), Δ-8-tetrahydrocannabinol (Δ8-THC), and Δ-9-tetrahydrocannabinolic acid (Δ9-THCA) using 6-point calibration curves of external cannabinoid standards relative to umbelliferone internal standard area (ranging in concentration from 3.00 × 10^−7^ M–6.40 × 10^−6^ M, Appendix A). For THCA compounds, only the Δ9-THCA standard was available; therefore, putative Δ8-THCA and Δ9-THCA concentrations could simply be interpreted as unresolved THCA. We also ran a standard of Δ9-THC, but it was not detected in any hemp samples (Appendix A), so it was not quantitated. Calibration curves were run once a day (*n* = 16) and response was calculated as the slope of the calibration curve forced through the origin (Appendix A). Calibration curves were applied to all samples run on the same day. Limits of quantitation (LOQ) and detection (LOD) were calculated from curve standard deviations and slope according to ICH guidelines [21]. Extract cannabinoid concentrations were then used to calculate dry mass concentrations in hemp samples. Missing dry mass values (*n* = 6) were imputed with the mean dry mass of all other samples (c.v. = 5.15%). There were no samples where cannabinoids were detected at concentrations below 3.00 × 10^−7^ M, but some CBN concentrations were below the method LOQ and are indicated in Appendix A as detected. Chromatograms of standards and characteristic samples can be seen in Appendix A.

### 3.4. Statistical Analysis

For all statistical models, plant chemistry was analyzed as the percent dry weight of specific metabolites, or as metabolite diversity based on relative percent dry weights. For diversity, we estimated the three most used hill numbers [22] (q = 0, richness; q = 1, Shannon; q = 2, Simpson) but reported q = 2 to emphasize common peaks while also including rarer compounds (results from other diversity measures are included in Appendix A). Hill numbers, when applied to spectroscopy data, account for the relative abundances of individual peaks. These are calculated based on their respective peak areas, which are normalized to proportional abundances. We used structural equation models (SEMs) to test a set of a priori causal hypotheses about relationships between cannabinoid synthesis (see Figure 1), plant varieties, water treatments (analyzed as water deficit versus control and flood versus control), untargeted phytochemical diversity, and arthropod diversity. The goal of our SEM modeling was to test causal hypotheses for suites of exogenous (manipulated or “predictor” variables), endogenous (unmanipulated variables that can be both “predictors” and “responses”), and latent (unmeasured) variables and to assess statistical support for a parsimonious model that prioritizes a priori hypotheses through the inclusion of key causal pathways. Model fits were confirmed with standard Chi-square statistics, with lower chi-square values (i.e., frequentist *p*-values higher than 0.4) being consistent with a good model fit. Alternative models with higher AIC values are included as R scripts in Appendix A. All parameter estimates and standard errors are included in path diagrams for a model that was the best fit; variances and z-scores for all estimates are provided in the form of R scripts in Appendix A. We also fit Bayesian multiple regressions simply for the heuristic purposes of showing posteriors of predicted values of z-scored chemistry response variables (CBD concentrations and overall phytochemical diversity) as a more flexible alternative to simple summary statistics. We set weakly informative priors for each of the regression coefficients, using normal distributions with a mean of 0 and a variance of 100, to center the estimates around a plausible null value; a precision parameter for these distributions was set as a diffuse gamma prior with shape and rate parameters of 0.01. All statistical analyses were executed in R version 4.2.2, with the lavaan package for SEM and rjags for Bayesian regression models. Information on priors, trace plots, convergence diagnostics, and model validations are included in the Appendix A. For reporting results, the emphasis of this study is on the ecological relevance and magnitude of effect sizes rather than statistical significance, in line with the consensus of the American Statistical Association [23,24].

## 4. Discussion

Our water stress experiments using two varieties of hemp contribute to basic phytochemical diversity theory and expand on a considerable body of knowledge on applied hemp chemistry, with a new focus on how chemical shifts can affect hemp arthropods. As studies accumulate, phytochemical diversity is clearly important in mediating plant–insect interactions, and the effects on insect communities are likely to be quite different in flowers versus leaves for *C. sativa* [10] (Fischedick et al. 2010) and other plants. The effects of both genetic (varieties) and environmental (wind and water stress) factors on hemp phytochemical diversity were expected, but it was interesting that while the Lifter variety had lower CBD than the Cherry Wine variety, Lifter plants were also characterized by higher phytochemical diversity. This shift is driven by higher cannabinoid concentrations in Cherry Wine varieties but lower overall LC-MS peak diversity, suggesting trade-offs in the biosynthesis of cannabinoids versus other specialized metabolites. These results add to what we know about genetic and environmental contributions to variation in plant-specialized metabolites [25,26,27]. For these two varieties of hemp in our experiments, the lack of variety–water-treatment interactions suggests that hemp’s metabolomic responses to water stress are not constrained by genetic factors.

We also found evidence that subtle shifts in hemp chemistry can cascade to affect arthropod diversity. Most of the research on *Cannabis* chemistry has focused on human physiological effects; thus, very little is known about how hemp chemistry affects arthropod communities. Our results do not support the emerging paradigm that phytochemical diversity can deter herbivory but increase overall arthropod diversity [2,3]. On hemp, the decreases in arthropod richness associated with increased phytochemical diversity included herbivores, predators, parasitoids, and omnivores. Focal work on the effects of chemical diversity as well as specific compounds on pest herbivores versus beneficial insects would likely yield a clearer view of the mechanisms by which chemistry affects arthropods. It is possible that increased chemical diversity may deter colonization or reduce the abundance of the types of generalized detrimental arthropods that are likely to colonize the introduced species. In our study, we did not investigate which plant organs arthropods were visiting or which were transient visitors, but these would be more powerful approaches to better understand what chemical profiles might be more attractive to detrimental versus beneficial arthropods, including those influenced by volatiles. While some terpenes are volatile compounds that likely contribute to the fragrance emitted by hemp inflorescences, this study did not measure plant headspace to quantify these emissions or explore their potential role in insect attraction.

In contrast to overall hemp chemistry, the effects of increased CBD levels did positively affect arthropod richness, which supports hypotheses that dominant herbivores are more negatively impacted, allowing for the colonization of a broader mix of herbivores and their enemies [2]. Water and wind stress further influenced these dynamics by indirectly affecting arthropod richness via plant size and CBD levels. For any experiments focusing on the bottom-up effects of plant resources or plant stress [28] on herbivore communities, it is important to consider causal models based on existing theory and empirical studies, since complex associations between perturbations like wind stress and responses such as plant size, phytochemistry, and arthropod diversity are likely. These complex pathways can yield effects that cancel each other out and create a lack of correlation (e.g., the pathway from flood to plant size to arthropod diversity is negative, while the pathway from flood to plant size to CBD to richness is positive). The changes in height likely altered plant architecture, which can affect arthropod diversity and interaction diversity, but the growth rate of the plant also affects both height and CBD content, causing a counteractive effect on arthropod diversity, accessibility, and, ultimately, community composition. These complex interactions warrant more investigation to understand mechanisms underlying putative causal pathways.

Using simple causal modeling, we found strong support for biochemical causal hypotheses suggested by the well-studied cannabinoid pathways, which is no surprise, but this approach could be used to test other hypothesized biochemical pathways. It was also interesting to see clear shifts in the compositions of cannabinoids based on variety and water stress, suggesting that these pathways are responsive to environmental cues.

In sum, our results provide important insights into the potential ecological consequences of hemp phytochemical diversity. These relationships could be important in hemp agricultural settings, where the goal is to maintain lower abundances of insects, such as the corn earworm (*Helicoverpa zea* Boddie) and the cannabis aphid (*Phorodon cannabis* Passerini), to minimize crop damage and maintain quality [29]. Conversely, in natural landscapes, feral hemp plants could serve as valuable resources for declining insect populations, perhaps providing pollen [30] or otherwise creating novel sources of habitat and food for diverse insect communities. These types of impacts could be particularly relevant for any regions in the world where hemp cultivation is expanding or where expansion is planned for the near future. Hemp chemistry comprises far more than just the cannabinoids that are the subject of extensive research, and it would be interesting to further examine redundancy hypotheses for hemp, such as the “entourage effect” and how it affects plant–arthropod interactions. Finally, in the context of climate change, understanding how environmental stresses shape these interactions will help with future agricultural management.

## Figures and Tables

**Figure 1 plants-14-00474-f001:**
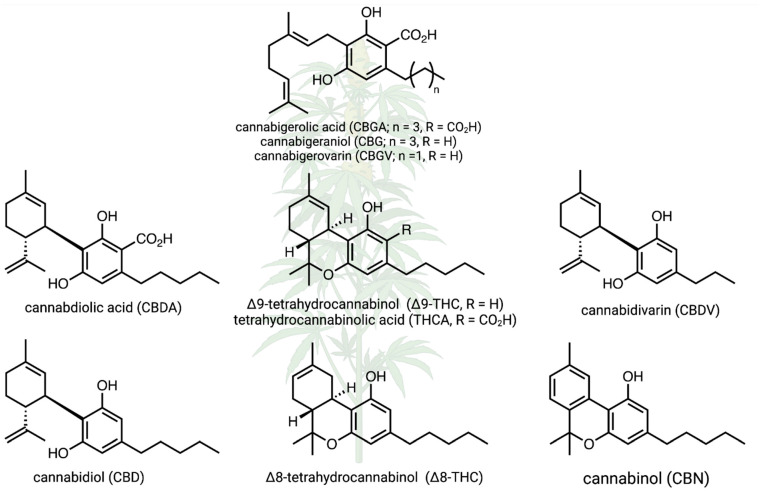
Canonical cannabinoid biochemical pathways in hemp, including only focal compounds. Several of these compounds are focal products in hemp research; here, we examined how environmental conditions and genetic variation affect this metabolism. Created in BioRender. Jeffrey, C. https://BioRender.com/z56q738 (accessed on 23 January 2025).

**Figure 2 plants-14-00474-f002:**
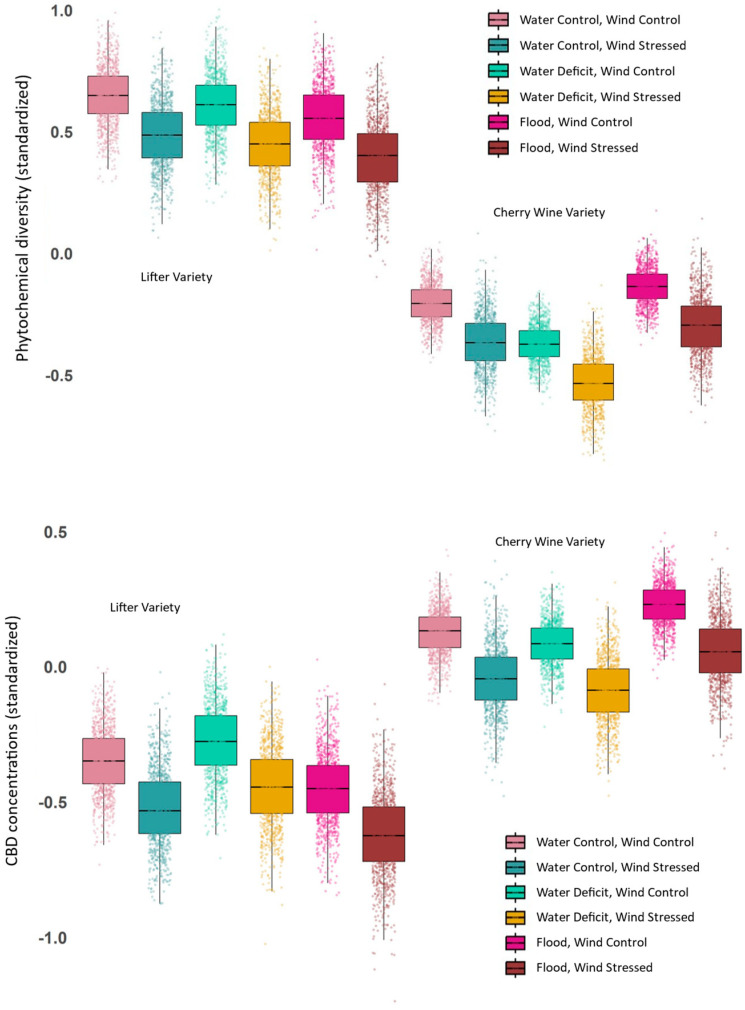
Bayesian estimates of medians and variance for standardized phytochemical diversity and CBD concentrations. Observed phytochemical diversity and CBD concentrations across hemp varieties and water treatments. Both response variables for standardized diversity (Simpson’s index, q = 2) are reported for Cherry Wine (C) and Lifter (L) clones under water deficit, control, and flood treatments. Variety had the largest effect on phytochemical diversity, with L plants showing consistently higher diversity. Stress effects (water deficit and wind stress) increased diversity across both varieties. Variation was estimated with a simple Bayesian approach, and boxplots and points are from a random sample of 1000 points from posterior distributions.

**Figure 3 plants-14-00474-f003:**
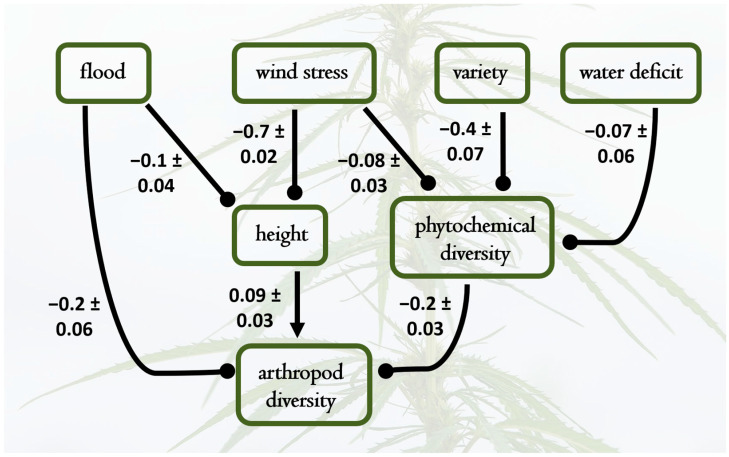
Best fit structural equation model results depicting causal relationships among water treatments, wind stress, plant variety, phytochemical diversity, plant height, and arthropod richness. Pathways not included had negligible path coefficients and contributed to poorer model fit. Standardized path coefficients are shown, with standard errors of individual effect sizes; the overall model had a good fit to the data (*X*^2^ = 6.4, DF = 7, *p* = 0.4). Notable causal hypotheses supported by the data were that shifting the hemp variety (from Lifter to Cherry Wine) yields lower phytochemical diversity, and in turn, higher phytochemical diversity causes lower arthropod richness. Increases in plant height positively influenced arthropod richness; thus, the negative effects of wind stress and flooding on plant height cascaded to indirectly decrease arthropod diversity.

**Figure 4 plants-14-00474-f004:**
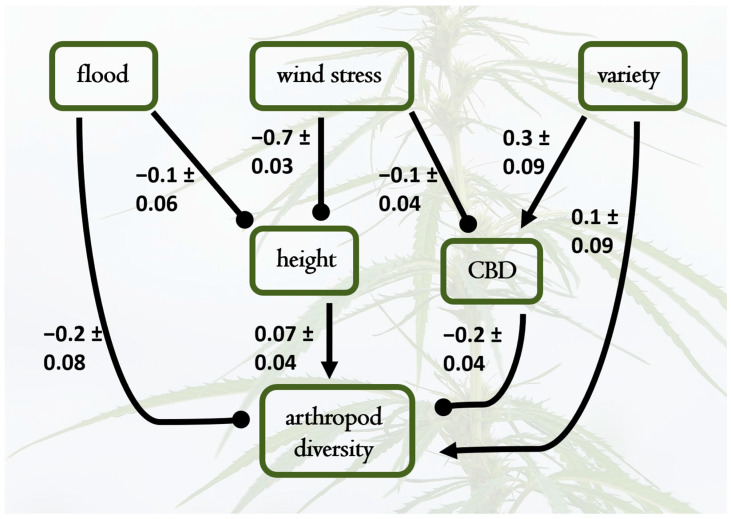
Best-fit structural equation model results depicting causal relationships among water treatments (flood and wind stress), plant height, plant variety (representing a shift from Lifter to Cherry Wine), CBD concentration, and arthropod diversity. Pathways not included had negligible path coefficients and contributed to poorer model fit. Standardized path coefficients are shown, with standard errors of individual effect sizes; the overall model was a good fit to the data (*X*^2^ = 6.4, DF = 7, *p* = 0.4). Notable supported causal hypotheses include positive effects of plant height on arthropod diversity, as well as direct negative effects of flooding and wind stress on plant height and CBD levels. Plant variety also influenced CBD concentrations (higher in the Cherry Wine variety), which subsequently caused lower arthropod diversity. The overall effects of variety on arthropod diversity are minimal because the indirect negative effects via increased CBD in Cherry Wine counteracted the direct positive effects of variety (shifting to Cherry Wine) on arthropod diversity.

**Figure 5 plants-14-00474-f005:**
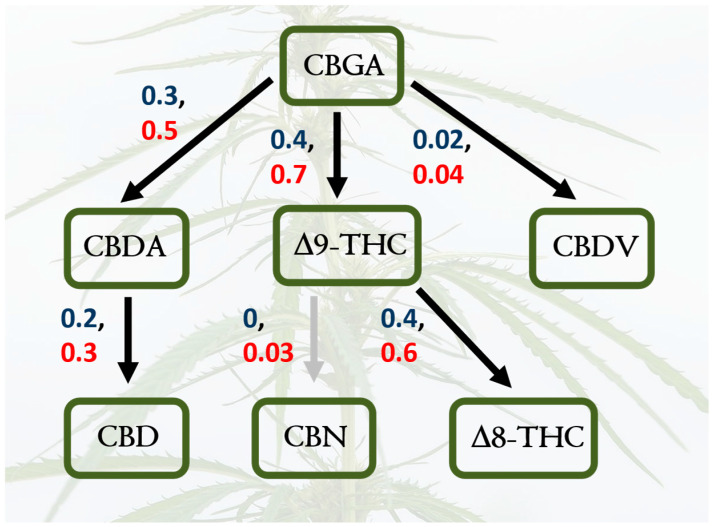
Shifts in cannabinoid pathways across hemp varieties and water treatments. The structural equation models support hypotheses of shifts in causal pathways for focal cannabinoids in response to varietal differences (Cherry Wine versus Lifter); path coefficients for the Cherry Wine variety are red, and coefficients for Lifter are blue. The grey arrow indicates a path coefficient that is no different from zero. Both models are good fits to the data (Appendix A).

## Data Availability

Data are available at: https://doi.org/10.5281/zenodo.14802874.

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
