# Peer review of "Effects of Water and Wind Stress on Phytochemical Diversity, Cannabinoid Composition, and Arthropod Diversity in Hemp"

_plants, 2025, doi:10.3390/plants14030474_

Round 1
Reviewer 1 Report
Comments and Suggestions for Authors
This report represents the revision of the article entitled “Effects of water and wind stress on phytochemical diversity, cannabinoid composition, and arthropod diversity in hemp”. The manuscript reports on the effects that some abiotic stress factors can have on plant growth and productivity, with a focus on Cannabis sativa and its major specialized metabolites (e.g. cannabinoids). The manuscript is interesting because the effects of stress conditions on the production of cannabinoids and their role have been insufficiently studied in the literature. This is particularly true for water stress, especially waterlogging and flooding. Although some work has been done under drought conditions, there is little data available for cases of excess water, which may also be due to climate change. The study of arthropods is also an interesting aspect of the work. In general, the manuscript is well written in English and a large amount of data has been collected. However, the data are not clearly presented. The discussion of the results is difficult to follow and the figures and tables are not described in detail in the text. I would completely revise the results and discussion section to better highlight the data collected and present the conclusions obtained more clearly. In general, I believe that the results obtained in this work say that the phytochemical diversity depends mainly on the varieties, but that there are also some differences in the varieties depending on the stress conditions; I believe that you should describe these aspects better, as it is confusing in this form. The same applies to the study of arthopods, for this reason, I think you should add the differences in arthopods between varieties to allow for comparison. Based on my observations, I would suggest a major revision.
You find below my comments on the manuscript:
Abstract:
- Line 13: please add the author to the scientific name of the plant: Cannabis sativa L.
Introduction:
- Line 28: what do you mean by “defensive redundancy”?
- Line 30: reword as “antiherbivore defense is one on the MOST focused area…”
- Line 45: please add the author to the scientific name of the plant, that must be in italics: Cannabis sativa L. This should be applied to all scientific names of plant and arthropods when they are cited for the first time in the manuscript. E.g. Helicoverpa zea at line 70 and others
- I would add in the introduction a short botanical description of C. sativa and the classification of the different types of C. sativa based on the chemical profile (you can find useful information here: dx.doi.org/10.1139/cjb-2023-0056).
Materials and methods:
- The chemical section with all the chemicals used (e.g. standards, solvents..) is missing
- I would integrate “appendix 1” text with the materials and methods into the main text, as it would be easier to read since some parts are now repeated. I would only include the figures and tables in the supplementary without the appendix heading. The only appendix I would leave in the supplementary is “appendix 4”, which should become “appendix 1”.
- Line 82: I believe this is figure S1.
- Line 82-84: some parts are not clear. Why do you refer to clones and not plants or varieties, strains, cultivars? What are they exactly? Where were the fabric pots stored? In a nursery or outdoors? Are the plants female or male? It is not clear how the 110 plants died before planting for wind stress if they were not planted. Why did you replace Trophy Wife with Lifter? I would completely reword this paragraph to include Appendix 1 and to make it clearer.
- Line 92: “At the end of the growing season” when exactly? How the inflorescences were dried?
- Line 108: the results should be presented only in the result section
- Line 109: Why did you use this extraction method? Can you add a reference? Also, why did you use methanol as extraction solvent and not acetone, which should be more effective for extracting cannabinoids?
- Line 112: which ionization method was used?
- Line 113: Please specify the gradient employed
- Line 116: In which acquisition mode were the peak areas analyzed? Scan mode, SIM, SRM?
- Line 117: How exactly do you estimate the phytochemical diversity? Did you calculate the sum of all areas or did you look at each area of each individual peak?
- Line 117-125: In which acquisition mode was the calibration curve prepared? Scan, SIM, SRM? Also, you should include a table with the figures of merit of all the calibration curves (e.g. linearity, equation, LOD, LOQ).
Results and discussion:
- It is important to add a representative profile of the analyzed extracts. You can add a figure in which you compare a profile of an extract from a control plant with the extracts from the other two conditions tested. You can do this for both clones/varieties analyzed. Please indicate in the profiles the compounds identified/supposed. It would be useful also to prepare a table with the information obtained by the chemical analysis (e.g. retention time, deprotonated molecule…)
- Figure 1: is incomplete (refer to https://doi.org/10.3389/fphar.2021.777804 ) and the cannabidivarin structure is incorrect (refer to https://pubchem.ncbi.nlm.nih.gov/compound/Cannabidivarin#section=2D-Structure).
- Figure 2: please add the unit to the y-axis. Did you perform an ANOVA to verify that the differences have a statistical significance? I would also discuss the differences in phytochemical diversity between the various treatments.
- Line 154: this is not clear to me, I would say that variety causes higher levels of phytochemical diversity compared to water and wind stress.
- Figure 3: It is not clear to me why you have not illustrated the effects of flooding on phytochemical diversity or the effects of variety and water deficit on height, but did show the effects of flooding on arthropod diversity. I think there is some information missing in this figure. The same is true for the other SEM images.
- Figure S2a,b: why are some arthropod orders and families repeated? Also, you have included three treatments in this paper (0.5, 1, and 2 gallons per hour), while the legend in the figure indicates four treatments (0.5, 1, 1.5, 2 gallons per hour). You should also add the unit information to the treatments. In Figure 2b, there is also a shift between the histogram bars and the family names. I would also delete for the histogram the orders/families with 0 frequency. Another important point is that the caption is missing. Figure S2 should also be discussed in detail in the results section, as it appears that some arthropods are only present in the stress treatments or in the control. It is important to describe all figures. I would also add a hystogram showing the differences in arthropod abundance in the two clones/varieties.
- Line 186: It is not true that Figure 3 “provided estimates of causal relationships between arthropod diversity, cannabinoids, flood, drought, wind stress, and variety”, it provides only the effects of flood and phytochemical diversity on arthropods. You should deeply revise this part also taking into account my previous comment on figure 3.
- Figure S4 and Table S2: please indicate the concentrations as ug/g or ug/mg. In addition, I think it would be important to indicate the statistical significance in the histogram bars and also in the data in Table S2. Furthermore, Figure S4 and Table S2 should be discussed in more detail in the results section.
- Figure S4: Since there is a difference between the varieties, I believe that the histograms and the discussion should be done separately for the two varieties.
- Table S2: The s.d. of CBDV is very high and is inconsistent with those reported in Figure S4
- Line 199: Table S3 is not present in the supplementary
- Figure 5 and Figure S3 are not clear to me. I do not understand how you can evaluate the biosynthetic pathway and the relations of each compounds without analyzing gene expression. I think this part is misunderstanding. I would therefore delete it unless you can discuss it in more detail and in a more understandable way.
- Line 223: the varieties analyzed are three
- I believe that the discussion should also be revised based on the revision of the results section.
References
- Some references cited in the text are missing
Author Response
We really appreciate the reviewer's thorough and professional feedback, which provided valuable insights into all aspects of our study and brought up some errors and typos. Their thoughtful comments and suggestions have significantly enhanced the clarity, rigor, and overall quality of the manuscript. Reviewer comments are included below in their entirety and responses are in italics.
Reviewer 1:
Line 13: please add the author to the scientific name of the plant: Cannabis sativa L.
We have added the author to the scientific name of the plant as requested.
Line 28: what do you mean by “defensive redundancy”?
We appreciate this comment, as the meaning of this term might not be clear to most readers. “Defensive redundancy” refers to the fact that most plants produce multiple secondary metabolites that may collectively contribute to defense, offering overlapping or complementary protection against a variety of natural enemies, such as herbivores and pathogens. We have added this clarification to the revision along with the appropriate references where it is carefully examined (Romeo et al. 2013, Richards et al. 2015).
Line 30: reword as “antiherbivore defense is one on the MOST focused area…”
Done.
Line 45: please add the author to the scientific name of the plant, that must be in italics: Cannabis sativa L. This should be applied to all scientific names of plant and arthropods when they are cited for the first time in the manuscript. E.g. Helicoverpa zea at line 70 and others
We appreciate this helpful suggestion and have updated the manuscript to include the author name for Cannabis sativa as Cannabis sativa L. and applied the same formatting to all scientific names of plants and arthropods when they are cited for the first time, including Helicoverpa zea Boddie.
I would add in the introduction a short botanical description of C. sativa and the classification of the different types of C. sativa based on the chemical profile (you can find useful information here: dx.doi.org/10.1139/cjb-2023-0056).
This is an excellent suggestion - we agree that including a brief botanical description and classification of C. sativa based on its chemical profile adds important context to the introduction. We have updated the manuscript accordingly.
The chemical section with all the chemicals used (e.g. standards, solvents..) is missing
We have completely rewritten the chemistry methods, including these details.
I would integrate “appendix 1” text with the materials and methods into the main text, as it would be easier to read since some parts are now repeated. I would only include the figures and tables in the supplementary without the appendix heading. The only appendix I would leave in the supplementary is “appendix 4”, which should become “appendix 1”.
We have modified the methods as suggested.
Line 82: I believe this is figure S1.
We have corrected this in the manuscript.
Line 82-84: some parts are not clear. Why do you refer to clones and not plants or varieties, strains, cultivars? What are they exactly? Where were the fabric pots stored? In a nursery or outdoors? Are the plants female or male? It is not clear how the 110 plants died before planting for wind stress if they were not planted. Why did you replace Trophy Wife with Lifter? I would completely reword this paragraph to include Appendix 1 and to make it clearer.
We appreciate these thoughtful questions and suggestions to improve the clarity of this section. Below, we address each question individually:
Why do you refer to clones and not plants, varieties, strains, or cultivars? What are they exactly?
We refer to them as clones because these plants were propagated from clonal cuttings rather than from seed; this is a common practice in cannabis cultivation to ensure genetic uniformity, and we added this clarification into the methods.
Where were the fabric pots stored? In a nursery or outdoors?
The fabric pots were stored outdoors in the field between growing seasons. These are like standard planting pots but made from fabric rather than plastic. There use is another common practice in cannabis cultivation. Soft pots are preferred in cannabis cultivation because they promote healthier root systems through improved aeration, prevent root circling with air-pruning, and enhance drainage for optimal plant growth. We have added these details to the revised methods.
Are the plants female or male?
This is a great question - this distinction is important and have amended the manuscript to clarify this. The updated text now reads: "One hundred-fifty female clones bred for CBD production were planted..."
How did the 110 plants die before planting if they were not planted?
The day before planting was scheduled, a wind event occurred while the plants were outdoors hardening off in their smaller pots. This event resulted in the death of 110 clones. Hardening off is a critical step to acclimate plants to outdoor conditions and prevent high mortality after planting. We have clarified this in the revised methods.
Why did you replace Trophy Wife with Lifter?
Due to the loss of clones, we needed a replacement quickly. Lifter clones were the only ones we could source within a reasonable driving distance, leaving us with no alternative options. We have added this detail to the revision.
Incorporating Appendix 1 and detailed information:
We have modified the manuscript, moving all methodological text from Appendix 1 to the main text.
Line 92: “At the end of the growing season” when exactly? How the inflorescences were dried?
We have updated the manuscript to include this information: “The second week of October, final inflorescence samples were collected and dried in brown paper bags under laboratory conditions to preserve their integrity for chemical analysis.”
Line 108: the results should be presented only in the result section
We agree and have not reported results here, only necessary methodological details.
Line 109: Why did you use this extraction method? Can you add a reference? Also, why did you use methanol as extraction solvent and not acetone, which should be more effective for extracting cannabinoids?
- Line 112: which ionization method was used?
- Line 113: Please specify the gradient employed
- Line 116: In which acquisition mode were the peak areas analyzed? Scan mode, SIM, SRM?
We appreciate these requests for clarification and have completely rewritten the chemistry methods, including all these details. Specifically, for the extraction method, we were following an Agilent white paper that was recommended by our instrument vendor which we have cited in the revision. In addition to cannabinoids, we were interested in more polar metabolites that may have been involved in plant-insect that could have been excluded by a less polar aprotic solvent such as acetone (such as flavonoid glycosides). We have had good success analyzing such compounds using methanol as an extraction solvent, and thought this approach would strike the right balance. We have specified in the revised methods that we used ESI and we have provided our gradient method. Finally, the MS indicates that peaks were analyzed in scan mode, but we have parenthetically added clarification that acquisition was carried out in TOF-only mode.
Line 117: How exactly do you estimate the phytochemical diversity? Did you calculate the sum of all areas or did you look at each area of each individual peak?
We estimated phytochemical diversity using Hill numbers, which are diversity equivalents of entropies (Jost 2006). Hill numbers account for the relative abundances of individual peaks, calculated based on their respective peak areas. This approach inherently considers the total peak area by normalizing individual peak areas to relative abundances, providing complete measures of diversity that incorporate both the number of detected compounds (richness) and their proportional abundances. So, this method avoids simply summing peak areas and instead ensures that diversity estimates reflect both the composition and relative contribution of individual compounds. We have clarified the methods for estimating diversity by adding the Jost citation and the above explanation with this addition to the methods: “Hill numbers, when applied to spectroscopy data, account for the relative abundances of individual peaks. These are calculated based on their respective peak areas, which are normalized to proportional abundances.”
Line 117-125: In which acquisition mode was the calibration curve prepared? Scan, SIM, SRM? Also, you should include a table with the figures of merit of all the calibration curves (e.g. linearity, equation, LOD, LOQ).
We have added clarification that the calibration curves were acquired using the same conditions as analytical samples. We have also provided calibration curve parameters in a table with other cannabinoid properties in the supplemental.
It is important to add a representative profile of the analyzed extracts. You can add a figure in which you compare a profile of an extract from a control plant with the extracts from the other two conditions tested. You can do this for both clones/varieties analyzed. Please indicate in the profiles the compounds identified/supposed. It would be useful also to prepare a table with the information obtained by the chemical analysis (e.g. retention time, deprotonated molecule…)
A supplemental figure was created showing characteristic profiles of cannabinoids in the two varieties and three water treatments, also the labeled standards profile is shown. Separate figures are given for CBD/CBDA and the other cannabinoids due to the extreme differences in concentrations. Molecular ion masses and retention times have been added to Table S2. We have also prepared a table with cannabinoid properties and calibration curve parameters.
Figure 1: is incomplete (refer to https://doi.org/10.3389/fphar.2021.777804 ) and the cannabidivarin structure is incorrect (refer to https://pubchem.ncbi.nlm.nih.gov/compound/Cannabidivarin#section=2D-Structure).
We appreciate the reviewer catching these issues and have corrected the error in the cannabidivarin structure. Additionally, we have clarified in the figure legend that the figure is not intended to represent a complete biosynthetic scheme but rather to focus on the specific compounds central to our analysis, which aligns with the scope of our study.
Figure 2: please add the unit to the y-axis. Did you perform an ANOVA to verify that the differences have a statistical significance? I would also discuss the differences in phytochemical diversity between the various treatments.
We have not added units to the y-axis because these are standardized values, so the units are simply standard deviations. We did not utilize ANOVA for several reasons, but primarily because these are descriptive results and we are not focused on frequentist hypothesis tests with this figure, rather just providing chemical profile summaries of the experimental plants with Bayesian estimates of central tendencies and dispersions. It is our general policy not to include frequentist statistical tests unless we had a priori planned that sort of inference from the onset. All our inferences come from the Structural Equation Models, but additional inferences by any reader or meta-analyst could be made from Bayesian estimates of central tendency and dispersion provided in Figure 2
Line 154: this is not clear to me, I would say that variety causes higher levels of phytochemical diversity compared to water and wind stress.
We agree and have made this clearer by editing as follows: "Stress responses caused by water deficit and wind damage were weaker than variety, but both forms of stress caused higher levels of phytochemical diversity." This wording highlights the fact that while variety has the strongest effect, water deficit and wind stress also contribute substantially to increased phytochemical diversity.
Figure 3: It is not clear to me why you have not illustrated the effects of flooding on phytochemical diversity or the effects of variety and water deficit on height, but did show the effects of flooding on arthropod diversity. I think there is some information missing in this figure. The same is true for the other SEM images.
We appreciate this insightful critique. The goal of SEM modeling is not to present a saturated model with all possible effect sizes, but rather to develop parsimonious models that adequately fit the data while prioritizing a priori hypotheses through the inclusion of key causal pathways. The effects mentioned by the reviewer were negligible and contributed to poorer model fit; thus, they were excluded from the final model to ensure clarity and accuracy in illustrating meaningful relationships. We have clarified this approach and rationale in the figure caption and have provided more details of our SEM approach in the methods. The full scripts included in the appendix allow for easy exploration of additional models and include comments.
Figure S2a,b: why are some arthropod orders and families repeated? Also, you have included three treatments in this paper (0.5, 1, and 2 gallons per hour), while the legend in the figure indicates four treatments (0.5, 1, 1.5, 2 gallons per hour). You should also add the unit information to the treatments. In Figure 2b, there is also a shift between the histogram bars and the family names. I would also delete for the histogram the orders/families with 0 frequency. Another important point is that the caption is missing. Figure S2 should also be discussed in detail in the results section, as it appears that some arthropods are only present in the stress treatments or in the control. It is important to describe all figures. I would also add a hystogram showing the differences in arthropod abundance in the two clones/varieties.
We appreciate these comments and recognize that confusion was caused because these figures were missing a caption, which has been added. There were only three levels of the water treatment. The repeating families were due to presenting data for two different varieties. These issues have been clarified in the figure captions and with new figures.
Line 186: It is not true that Figure 3 “provided estimates of causal relationships between arthropod diversity, cannabinoids, flood, drought, wind stress, and variety”, it provides only the effects of flood and phytochemical diversity on arthropods. You should deeply revise this part also taking into account my previous comment on figure 3.
We really appreciate this concern, but the statement that Figure 3 “provides estimates of causal relationships between arthropod diversity, cannabinoids, flood, drought, wind stress, and variety” is accurate within the context of the SEM approach, which focuses on the estimation of all these causal relationships. However, only meaningful and supported pathways are included in the figure. Relationships not shown in Figure 3 can be assumed to be negligible or close to zero. This approach has been clarified in the figure captions, aligning with the focus on presenting parsimonious and interpretable models.
Figure S4 and Table S2: please indicate the concentrations as ug/g or ug/mg. In addition, I think it would be important to indicate the statistical significance in the histogram bars and also in the data in Table S2. Furthermore, Figure S4 and Table S2 should be discussed in more detail in the results section.
The emphasis of this study is on the ecological relevance and magnitude of effect sizes rather than statistical significance, in line with the consensus of the American Statistical Association (ASA) in 2016 (10.1080/00031305.2016.1154108). The ASA has emphasized that reliance on statistical significance can lead to misinterpretation and overemphasis on arbitrary thresholds, and subsequent papers continue to bolster this advice (e.g., Wasserstein et al., 2019; DOI: 10.1080/00031305.2019.1583913). Instead, we focus on presenting effect sizes with measures of uncertainty (e.g., credible intervals) to provide a more relevant interpretation of the data and its ecological importance. While statistical significance is not indicated anywhere in the manuscript, all relevant effect sizes and their uncertainty are presented to facilitate interpretation. This approach allows readers to evaluate the importance of the results without the limitations imposed by p-value thresholds. We have added this justification and references to the statistics section of the methods.
Figure S4: Since there is a difference between the varieties, I believe that the histograms and the discussion should be done separately for the two varieties.
This is a great suggestion, and we have made this change to the figure and highlight the difference between varieties in the discussion. There is no interaction between variety and other endogenous variables, so we do not separate the effects of other manipulated factors by variety.
Table S2: The s.d. of CBDV is very high and is inconsistent with those reported in Figure S4
Table S2 and Figure S4 are reporting different data. Table S2 does not include wind effects, and as such the high standard deviation is reflecting large differences in CBDV between wind stressed and unstressed plants. Also, table S2 separates data into each varietal whereas Figure S4 includes all varietals within each treatment group. Figure S4 error bars report standard error and not standard deviation, so the higher n resulting from combining all varietals together leads to these standard error bars looking much smaller than the standard deviation (s.e. = s.d./sqrt(n)).
Line 199: Table S3 is not present in the supplementary
We really appreciate the reviewer catching this mistake, this was meant to reference Appendix 4; we have corrected this in the revision.
Figure 5 and Figure S3 are not clear to me. I do not understand how you can evaluate the biosynthetic pathway and the relations of each compounds without analyzing gene expression. I think this part is misunderstanding. I would therefore delete it unless you can discuss it in more detail and in a more understandable way.
We appreciate the reviewer's comments and acknowledge that our causal modeling approach was not clearly communicated in the original manuscript. The focus of this analysis is not on directly measuring gene expression or genetic mechanisms underlying cannabinoid biosynthesis, but rather on testing well-established causal hypotheses about these biosynthetic pathways using structural equation modeling (SEM). This approach is particularly powerful for assessing causal relationships in systems where a strong theoretical and empirical framework exists, such as the well-studied cannabinoid biosynthetic pathways. SEM allows us to evaluate and compare the relative importance of different causal pathways (represented as arrows in the models) across conditions, whether those conditions are experimental or observational, such as between the two hemp varieties (Cherry Wine and Lifter). These models are informed by prior knowledge of cannabinoid biosynthesis and are designed to evaluate how environmental factors, like water treatments, interact with varietal differences to influence cannabinoid production. While gene expression analysis would provide excellent insights, it is not required to test the causal hypotheses presented here, as these hypotheses are based on known relationships between cannabinoids and their biosynthetic precursors. To address potential confusion, we have revised the figure caption to clarify that the "genetic effects" referenced in the earlier version refer to varietal differences rather than direct genetic mechanisms. Additionally, we have added text to the manuscript to explain the utility of SEM for testing causal hypotheses in this context, even in the absence of direct genetic manipulations. Revised figure captions:
Figure 5. Shifts in cannabinoid pathways across hemp varieties and water treatments. The structural equation models support hypotheses of shifts in causal pathways for focal cannabinoids in response to varietal differences (Cherry Wine versus Lifter); path coefficients for the Cherry Wine variety are red and coefficients for Lifter are blue. Both models are good fits to the data (Appendix 4).
Figure S3. Best fit Structural equation models (SEM) depicting putative causal relationships among focal cannabinoids. SEM allowed us to evaluate and compare the relative importance of different causal pathways (represented as arrows in the models) across conditions. These models were informed by prior knowledge of cannabinoid biosynthesis (Figure 1) and were designed to evaluate how water treatments influence cannabinoid concentrations. The SEMs support hypotheses of shifts in causal pathways for cannabinoids in response to the different levels of water treatment.
Line 223: the varieties analyzed are three
We have updated the manuscript to accurately reflect that two (not three) varieties were analyzed.
I believe that the discussion should also be revised based on the revision of the results section.
Points above about the results were due to omissions in the way the results were reported. The responses above with associated revisions should address this concern. Still, we have edited the discussion with all reviewer concerns in mind and have tracked changes throughout.
Some references cited in the text are missing
We have carefully checked the references and added the previously missing Park et al. and Caplan et al. references.
Reviewer 2 Report
Comments and Suggestions for Authors
To understanding how extreme weather events affect phytochemical diversity is an important part of predicting responses of plant-arthropod interactions to climate change. This manuscript investigated the effects of water and wind stress on phytochemical diversity, cannabinoid composition, and arthropod diversity in hemp, an important medicinal and economic plant. The discoveries of present study are interesting, and the research results would be helpful to maintain lower abundances of harmful insects, and minimize crop damage and maintain quality of hemps. This manuscript is written well, but there are still some issues that need to be addressed.
P82: Figure S2 should be corrected to Figure S1.
P85: Was cannabinoid fragrant substance? Beside cannabionoids type of compounds, how about the change in fragrance emitted by hemp’s inflorescences? These fragrance would be attractive to insects.
P109: Fisher Optima methanol is suggested to be replaced by methanol.
P111: uL should be corrected to μL.
P119: abbreviation of Δ9-THCA should be same with Δ9-THC in Figure 1.
Please provide the TIC figure of hamp. Could it differentiate Δ8-THC and Δ9-THC?
P124: Question on “Missing dry mass values were imputed with mean dry mass of all other samples (c.v. = 5.15%)” : Why some of the dry mass values could be missed? Is it correct to use the mean dry mass of all other samples to replace the missing value? Since each cannabionoid concentration was calculated by dry weight, please add the method for water content detection, especially in fragrant inflorescence of hemp.
P162, Figure 2: Please add X and Y axis in Figure 2. There are totally 150 clones for these six groups as indicated in Table S1, and the number of plants in the range of 5-31 in each group. How many hamp’s inflorescences samples in each group did you detect, and why there are so many points (1000) in each group?
P165, Figure 2: Phytochemical diversity are different in C and L. In detail, which kinds of metabolites differ in these two clones by non-target metabolites analysis? Please added TIC diagrams in HPLC-MS analysis, especially the representative C and L clones, and provide some representative data in Supplementary materials. For CBD calculation by HPLC/MS method, please provide diagrams of standards and the representative samples, calibration equation for each CBD and some representative data for each CBD in these six groups.
P170, Figure 3: The data in Figure 3 are expressed so strange, should 0 be added before the dot? So did Figure 4 and 5.
P192: 0.09 or 0.08? please check Figure 3.
Supplementary:
(1) Figure S2a and 2b: The letters in Figure 2a and 2b are quite small. Please add notes to differentiate two varieties Cherry Wine clones (Left) and Lifter clones in these figures.
(2) Figure S4: Please add the unit of concentration in each Figure. The unit of μM/g seems quite strange, could you change it to commonly used units, such as %, μg/g or mg/g? ppm, or ppt (ng/kg) as you described in Table S2? The data units in Figure S4 and Table S2 should be same.
(3) There is no Table S3 to define the effects of other compounds on arthropod richness and diversity?
(4) Please added notes for Figure S1-S3.
Author Response
We really appreciate the reviewer's thorough and professional feedback, which provided valuable insights into all aspects of our study and brought up some errors and typos. Their thoughtful comments and suggestions have significantly enhanced the clarity, rigor, and overall quality of the manuscript. Reviewer comments are included below in their entirety and responses are in italics.
REVIEWER 2
P82: Figure S2 should be corrected to Figure S1.
We thank the reviewer for catching this and have corrected the error.
P85: Was cannabinoid fragrant substance? Beside cannabionoids type of compounds, how about the change in fragrance emitted by hemp’s inflorescences? These fragrance would be attractive to insects.
This is an interesting and helpful question. Some terpenes are volatile and fragrant compounds, and their concentrations likely contribute to the overall fragrance emitted by hemp inflorescences. However, we did not measure the headspace of the plants to directly quantify or analyze their fragrance in this study. While differences in the diversity of these volatile compounds may correlate with fragrance and could potentially influence insect attraction, exploring this relationship is beyond the scope of this paper. We appreciate this comment and agree that this would be a valuable area for future research. We added the following: “While some terpenes are volatile compounds that likely contribute to the fragrance emitted by hemp inflorescences, this study did not measure plant headspace to quantify these emissions or explore their potential role in insect attraction.”
P109: Fisher Optima methanol is suggested to be replaced by methanol.
Done.
P111: uL should be corrected to μL.
We have updated all instances of "uL" to "μL" throughout the manuscript to ensure correct and consistent usage.
P119: abbreviation of Δ9-THCA should be same with Δ9-THC in Figure 1.
Please provide the TIC figure of hamp. Could it differentiate Δ8-THC and Δ9-THC?
We have included a supplemental figure showing characteristic chromatograms of standards and samples from each experimental group. The standard chromatograms show that we can differentiate Δ8-THC and Δ9-THC and Δ9-THC was not detected in any samples. Due to the overwhelming concentration of CBD and CBDA in these samples, TICs are not very informative, so EICs of quantified cannabinoids were included. The figure 1 had read “Δ-8-tetrahydrocannabinol (Δ8-THC), and Δ-9-tetrahydrocannabinol (Δ9-THCA)" so we have changed it to read Δ9-THC.
P124: Question on “Missing dry mass values were imputed with mean dry mass of all other samples (c.v. = 5.15%)” : Why some of the dry mass values could be missed? Is it correct to use the mean dry mass of all other samples to replace the missing value? Since each cannabionoid concentration was calculated by dry weight, please add the method for water content detection, especially in fragrant inflorescence of hemp.
We utilized completely dried plant material as mentioned in the methods.
P162, Figure 2: Please add X and Y axis in Figure 2. There are totally 150 clones for these six groups as indicated in Table S1, and the number of plants in the range of 5-31 in each group. How many hamp’s inflorescences samples in each group did you detect, and why there are so many points (1000) in each group?
We have not added units to the y-axis because these are standardized values, so the units are simply standard deviations; the legend provides all the information that would be conveyed in an x-axis. The points in Figure 2 represent samples drawn from the posterior distributions generated through our Bayesian statistical analysis, where 1000 samples were taken to characterize the uncertainty and variability in the parameter estimates for each group. The use of posterior sampling is a standard practice in Bayesian modeling, as it provides a robust way to illustrate the range of plausible values for the parameters. The number of data points in Figure 2 reflects the posterior sampling process rather than the number of individual inflorescences or plants. We clarified this in the figure legend to clarify this point and avoid any potential misunderstanding.
P165, Figure 2: Phytochemical diversity are different in C and L. In detail, which kinds of metabolites differ in these two clones by non-target metabolites analysis? Please added TIC diagrams in HPLC-MS analysis, especially the representative C and L clones, and provide some representative data in Supplementary materials. For CBD calculation by HPLC/MS method, please provide diagrams of standards and the representative samples, calibration equation for each CBD and some representative data for each CBD in these six groups.
We have provided a supplemental figure with representative chromatograms of cannabinoids in each experimental group and a table containing calibration curve parameters. We have included an updated supplemental table containing ppt cannabinoid concentrations for each experimental group. Due to the overwhelming concentration of CBD and CBDA in these samples, TICs are not very informative, so EICs of quantified cannabinoids were included.
P170, Figure 3: The data in Figure 3 are expressed so strange, should 0 be added before the dot? So did Figure 4 and 5.
We have made this change.
P192: 0.09 or 0.08? please check Figure 3.
We have fixed this typo.
- Figure S2a and 2b: The letters in Figure 2a and 2b are quite small. Please add notes to differentiate two varieties Cherry Wine clones (Left) and Lifter clones in these figures.
We have updated the figure to address this helpful comment.
(2) Figure S4: Please add the unit of concentration in each Figure. The unit of μM/g seems quite strange, could you change it to commonly used units, such as %, μg/g or mg/g? ppm, or ppt
(ng/kg) as you described in Table S2? The data units in Figure S4 and Table S2 should be same.
We have updated the figure to address this helpful comment.
(3) There is no Table S3 to define the effects of other compounds on arthropod richness and diversity?
The reference to this table was an error, as this information is available in the scripts in Appendix 4. We have referenced this appendix in the revision.
(4) Please added notes for Figure S1-S3.
We have added figure captions.
Reviewer 3 Report
Comments and Suggestions for Authors
In the current article “Effects of water and wind stress on phytochemical diversity, cannabinoid composition, and arthropod diversity in hemp” the author has reported that both genetic variety and environmental stress have substantial effects on variation in hemp phytochemical diversity and cannabinoid composition, and these effects cascaded to alter the arthropod communities on flowers. There are several huge questions that need to be answered including cosmetics on Figures.
1. Similar work has been reported previously by Park et al. Journal of Cannabis Research (2022) 4:1 “Effects of short-term environmental stresses on the onset of cannabinoid production in young immature flowers of industrial hemp (Cannabis sativa L.)”. Adding the effects on anthropod communities might be interesting. This should be focus in the abstract and introduction majorly. Furthermore the inside discussion also majorly focusing the environmental stress effects on phytochemical diversity only.
2. It would be great if author can give more comparison idea about potential beneficial insect vs detrimental insect diversity?
3. What organ the insects/anthropod were visiting?
4. In the data, Insects were staying or just visiting on the hemp plant?
5. If a comparison between visiting frequencies over flower vs lower parts can be provided, the article should be termed as significant.
6. In figure 1. Shows CBV, while the footnote says CBDV?
7. Page 4 &7, line 54 & 117, Is it THCA or THC; correct the Figure 1 and other places accordingly?
8. Figure S1 doesnot look appropriate after manual additions, need to be corrected.
9. Figure 2 in manuscript and Appendix 2 is difficult to read, need to reconstruct.
10. In figure S4 the water treatments and wind stress should be clearly define along with writing in the footnote.
Author Response
We really appreciate the reviewer's thorough and professional feedback, which provided valuable insights into all aspects of our study and brought up some errors and typos. Their thoughtful comments and suggestions have significantly enhanced the clarity, rigor, and overall quality of the manuscript. Reviewer comments are included below in their entirety and responses are in italics.
REVIEWER 3
- Similar work has been reported previously by Park et al. Journal of Cannabis Research (2022) 4:1 “Effects of short-term environmental stresses on the onset of cannabinoid production in young immature flowers of industrial hemp (Cannabis sativa L.)”. Adding the effects on anthropod communities might be interesting. This should be focus in the abstract and introduction majorly. Furthermore the inside discussion also majorly focusing the environmental stress effects on phytochemical diversity only.
This is a really good point, and we acknowledge the relevance of the work by Park et al. (2022), which we had already cited in our original manuscript (line 40 – it had been omitted from the references list, which we have corrected). That study provides valuable insights into the effects of environmental stresses on the concentrations of major cannabinoids. In contrast, our study emphasizes phytochemical diversity and its cascading effects, including impacts on arthropod communities, which adds a complementary perspective. To address this comment, we have revised the manuscript to more explicitly highlight the effects of environmental stress on arthropod communities, ensuring that this aspect is given appropriate focus in the abstract and introduction (the discussion already included a strong focus on cascading effects on arthropods).
- It would be great if author can give more comparison idea about potential beneficial insect vs detrimental insect diversity?
We appreciate this suggestion and acknowledge the value of distinguishing between beneficial and detrimental insect diversity when considering the ecological impacts of hemp phytochemistry. While our current analysis focuses on the overall arthropod community, this approach aligns with the primary goals of the study, which emphasize broader patterns of diversity and community-level interactions. However, we recognize the importance of exploring the dynamics of beneficial and detrimental groups and agree that this could provide valuable insights in future research. To address this point, we have modified parts of the discussion, including this: “Focal work on the effects of chemical diversity as well as specific compounds on pest herbivores versus beneficial insects would likely yield a clearer view of mechanisms by which chemistry affects arthropods. It is possible that increased chemical diversity may deter colonization or reduce the abundance of the types of generalized detrimental arthropods that are likely to colonize introduced species.” We also added this to the Discussion: “In our study, we did not investigate which plant organs arthropods were visiting or which were transient visitors, but these would be more powerful approaches to better understand what chemical profiles might be more attractive to detrimental versus beneficial arthropods, including those influenced by volatiles. While some terpenes are volatile compounds that likely contribute to the fragrance emitted by hemp inflorescences, this study did not measure plant headspace to quantify these emissions or explore their potential role in insect attraction.”
- What organ the insects/anthropod were visiting?
We did not specifically record which plant organs the insects or arthropods were collected from during sampling. Our methodology was focused on capturing a representative sample of the arthropod community associated with the hemp plants as a whole, rather than focusing on interactions with specific plant organs. While this level of detail was outside the scope of the current study, we agree that it could provide valuable insights into plant-insect interactions in future research. To address this, we have added the following sentence to the Discussion: " In our study, we did not investigate which plant organs arthropods were visiting or which were transient visitors, but these would be more powerful approaches to better understand what chemical profiles might be more attractive to detrimental versus beneficial arthropods, including those influenced by volatiles. While some terpenes are volatile compounds that likely contribute to the fragrance emitted by hemp inflorescences, this study did not measure plant headspace to quantify these emissions or explore their potential role in insect attraction."
- In the data, Insects were staying or just visiting on the hemp plant?
It is possible that some of the insects observed were transient visitors rather than being specifically associated with the hemp plants. However, the insects included in the data were collected directly from the plants using beat sheets, aspirators, and visual inspection, ensuring that they were physically present on the plants at the time of sampling. Additionally, most of the collected insects were identified as taxa generally associated with hemp. We appreciate the opportunity to clarify this point. To address this, we have added the following sentence to the discussion: "In our study, we did not investigate which plant organs arthropods were visiting or which were transient visitors, but these would be more pwerful approaches to better understand what chemical profiles might be more attractive to detrimental versus beneficial arthropods, including those influenced by volatiles. While some terpenes are volatile compounds that likely contribute to the fragrance emitted by hemp inflorescences, this study did not measure plant headspace to quantify these emissions or explore their potential role in insect attraction."
- If a comparison between visiting frequencies over flower vs lower parts can be provided, the article should be termed as significant.
We agree that comparing visiting frequencies between flowers and lower plant parts would be an important and valuable component to include. However, we did not record specific information on which plant organs were visited or where on the plant insects were collected from during our sampling.
As this level of detail was outside the scope of our study, we are unable to provide such a comparison in the current manuscript. The additional sentences outlined above acknowledge this shortcoming.
- In figure 1. Shows CBV, while the footnote says CBDV?
We have fixed this typo.
- Page 4 &7, line 54 & 117, Is it THCA or THC; correct the Figure 1 and other places accordingly?
We appreciate the reviewer catching these issues and have corrected the ambiguities throughout.
- Figure S1 does not look appropriate after manual additions, need to be corrected.
We have corrected errors in the figure.
- Figure 2 in manuscript and Appendix 2 is difficult to read, need to reconstruct.
We have not added units to the y-axis because these are standardized values, so the units are simply standard deviations; the legend provides all the information that would be conveyed in an x-axis. The points in Figure 2 represent samples drawn from the posterior distributions generated through our Bayesian statistical analysis, where 1000 samples were taken to characterize the uncertainty and variability in the parameter estimates for each group. The use of posterior sampling is a standard practice in Bayesian modeling, as it provides a robust way to illustrate the range of plausible values for the parameters. The number of data points in Figure 2 reflects the posterior sampling process rather than the number of individual inflorescences or plants. We clarified this in the figure legend to clarify this point and avoid any potential misunderstanding. Figure S2 has been modified for easier viewing and interpretation.
- In figure S4 thewater treatments and wind stress should be clearly define along with writing in the footnote.
Figure S4 has been modified for easier viewing and interpretation.
Reviewer 4 Report
Comments and Suggestions for Authors
General
Unfortunately, I've been overambitious. After reading the ms, I realize that I don't seize the statistical procedures, which makes a meaningful review impossible.
Therefore, I can't answer many questions asked by the journal.
Figure 2. When it comes to the most basic information, which can also be instructive: is there any mention of sample size? My reading was maybe too superficial, but Adobe Acrobat does not find occurrences of "replicates" or "n values", either.
The graph is pretty, but what do the points mean ("a random sample of 1000 points from posterior distributions")?
Figures 3-5 are very similar, they could easily be condensed into one figure. The box plots don't need to be centered on the plant, just as well to show 3 plots side-by-side, this would even make it much easier to compare. Plant background image could be scaled to cover all 3 plots, a magnified picture could even be apealling froma graphical point of view.
"Structural equation models" sounds interesting, it's just that I don't have a clue what it is.
What I expected to find here and in Figure 2 is info on significant differences (term "significant" also not found in document), the models could be added on top of the very basic information.
Author Response
We really appreciate the reviewer's thorough and professional feedback, which provided valuable insights into all aspects of our study and brought up some errors and typos. Their thoughtful comments and suggestions have significantly enhanced the clarity, rigor, and overall quality of the manuscript. Reviewer comments are included below in their entirety and responses are in italics.
REVIEWER 4
Figure 2. When it comes to the most basic information, which can also be instructive: is there any mention of sample size? My reading was maybe too superficial, but Adobe Acrobat does not find occurrences of "replicates" or "n values", either.
The graph is pretty, but what do the points mean ("a random sample of 1000 points from posterior distributions")?
The points in Figure 2 represent samples drawn from the posterior distributions generated through our Bayesian statistical analysis, where 1000 samples were taken to characterize the uncertainty and variability in the parameter estimates for each group. The use of posterior sampling is a standard practice in Bayesian modeling, as it provides a robust way to illustrate the range of plausible values for the parameters. The number of data points in Figure 2 reflects the posterior sampling process rather than the number of individual inflorescences or plants. We clarified this in the figure legend to clarify this point and avoid any potential misunderstanding. The number of replicates and sample size are in table S1.
Figures 3-5 are very similar, they could easily be condensed into one figure. The box plots don't need to be centered on the plant, just as well to show 3 plots side-by-side, this would even make it much easier to compare. Plant background image could be scaled to cover all 3 plots, a magnified picture could even be apealling froma graphical point of view.
"Structural equation models" sounds interesting, it's just that I don't have a clue what it is.
We appreciate the suggestion to combine Figures 3-5 into a single figure to facilitate easier comparison. However, these figures represent distinct structural equation models (SEMs), each visualizing different variables and addressing specific research questions. For example, Figure 3 focuses on phytochemical diversity, Figure 4 examines CBD concentration, and Figure 5 highlights cannabinoid pathways. Combining them into one figure would compromise the clarity and interpretability of the models, as each figure is designed to emphasize unique aspects of the study. We recognize that SEMs may not be familiar to all readers, and we have ensured the text includes a clear explanation of their purpose and how they test causal hypotheses based on established biosynthetic pathways. In the revision, we have added more details in the methods and figure captions about how SEM works.
What I expected to find here and in Figure 2 is info on significant differences (term "significant" also not found in document), the models could be added on top of the very basic information.
The emphasis of this study is on the ecological relevance and magnitude of effect sizes rather than statistical significance, in line with the consensus of the American Statistical Association (ASA). The ASA has emphasized that reliance on statistical significance can lead to misinterpretation and overemphasis on arbitrary thresholds, and subsequent papers continue to bolster this advice (e.g., Wasserstein et al., 2019; DOI: 10.1080/00031305.2019.1583913). Instead, we focus on presenting effect sizes with measures of uncertainty (e.g., credible intervals) to provide a more nuanced interpretation of the data and its ecological importance. While statistical significance is not indicated in Figure 2 or elsewhere, all relevant effect sizes and their uncertainty are presented to facilitate interpretation. This approach allows readers to evaluate the importance of the results without the limitations imposed by p-value thresholds. We have clarified this in the revision and added the above statements and references.
Round 2
Reviewer 1 Report
Comments and Suggestions for Authors
I thank the authors for their responses to my comments, I think the manuscript has improved. However, I still have some comments and suggestions, so I suggest a minor revision.
A first general remark about the figures, the figures, especially those in the supplementary, must be renumbered as they are cited in the manuscript. Some examples are reported below.
-Line 49: I would specify that the plant analyzed in this work is the fiber-type. “Here, we examine the effects of water availability on the phytochemical profiles of agricultural fiber-type Cannabis sativa L.
-Line 93: please check the spelling of Cannabis
-Line 99: it is still not clear to me when you planted the clones, on July 1 or July 9? Also I would reword this sentence as “During the hardening off period, prior to planting in the bigger pots, a wind event on 27-June killed 110 of the 150 clones and they were replaced.”
-Line 101: So only Cherry Wine clones survived after the wind event? If this is correct, I would rephrase this part as: "Only Cherry Wine varieties survived the wind event and the Trophy Wife variety was replaced by Lifter due to its availability. While a fully factorial design was maintained, treatment representation was unbalanced with two hemp varieties: 108 Cherry Wine (C) clones and 42 Lifter (L) clones. "
-Line 94: I would reword as follow: “The pots were filled to 80% of capacity with a 1:1 ratio mixture of SOAR Potting Mix and Tahoe Propagation Mix (Full Circle Compost), mulched with rice straw and were stored outdoors in the field.”
-Lines 142-150: This part has already been repeated, please check the section and remove the repetitions.
-Line 164: please remove “(TOF only)
-Line 169: I would reword as: “Peaks, analyzed in scan mode, with height greater than 20,000 counts were extracted, and compounds found to have peak height greater than 100,000 in fewer than two individuals were removed.”
-Line 171: I would reword this sentence as: “Calibration curves were generated in scan mode for cannabinoid standards using the same LC-MS conditions described above.”
-Line 172: the term “quantified” is repeated
-Table S3: check the spelling of “cannabinoid”. How did you calculated the parameters of the calibration curve of Δ8-THC in the absence of the commercial standard? I would remove the Calibration Curve Parameters of this compound.
- I believe that Table S2 and S3 should be interchanged, since Table S3 is quoted in the text before Table S2.
-Line 176, Table S2 and Figure S4: I believe it would be better to reword this sentence as “For THCA compounds, only the Δ9-THCA standard was available; therefore, putative Δ8-THCA concentrations could semi-quantified based on Δ9-THCA calibration curve”
-Line 176: (3.00x10-7 M – 6.40x10-6 M) what this refers to?
-Line 192: can you add in the supplementary a representative MS profile with all the compounds considered for the hill numbers test?
- Figure 1: the structures of delta9-Tetrahydrocannabinol, delta8-Tetrahydrocannabinol and cannabinol are also incorrect
-Figure S2 is cited in the text after Figure S5, please revise the figures number.
-Figure S2: please add a heading to the graphs so it is clear whether it is Cherry Wine or Lifter. Also the treatment legend is still unclear, please provide only the three conditions in the legend. The third and fourth graphs are missing the first family name, and some families are still repeated, e.g. Coccinelidae.
-Figure S4 and Table S2: I still believe that you should indicate the concentrations as ug/kg instead of ppt and that the statistical significance (e.g. ANOVA) in this case is important, especially in the figure, to evaluate the differences among the treatments.
-Figure S4: Lifter clones were not exposed to wind stress, how is it possible that the hatched bars are also displayed for Lifted clones? Also replace the word hashed with hatched.
-Based on you reply “Table S2 and Figure S4 are reporting different data. Table S2 does not include wind effects, and as such the high standard deviation is reflecting large differences in CBDV between wind stressed and unstressed plants. Also, table S2 separates data into each varietal whereas Figure S4 includes all varietals within each treatment group. Figure S4 error bars report standard error and not standard deviation, so the higher n resulting from combining all varietals together leads to these standard error bars looking much smaller than the standard deviation (s.e. = s.d./sqrt(n)).” I believe that you should specify these details in the captions.
-Line 275: you can compare these results with the quantitative data (Figure S4) and the information about the arthropod (Figure S2) obtained in this work. CBD is higher in Cherry Wife variety, does this variety host a higher arthropod diversity? Furthermore, wind stress in Figure S4 does not seem to have a negative impact on CBD content, while Figure 4 seems to suggest this trend. How can this aspect be explained?
-Line 302: I do not think it's true that Lifter variety has a higher CBD content than Cherry Wine. Also, Cherry Wine is characterized by a lower phytochemical diversity as stated in the results section.
Author Response
We appreciate the attention to details from all the reviewers and have improved the manuscript again based on their suggestions.
-Line 49: I would specify that the plant analyzed in this work is the fiber-type. “Here, we examine the effects of water availability on the phytochemical profiles of agricultural fiber-type Cannabis sativa L.
We agree this is an important detail, however, the hemp analyzed in this study was not bred for fiber production. We have specified that it was bred for CBD production (L 92).
-Line 93: please check the spelling of Cannabis
Corrected.
-Line 99: it is still not clear to me when you planted the clones, on July 1 or July 9? Also I would reword this sentence as “During the hardening off period, prior to planting in the bigger pots, a wind event on 27-June killed 110 of the 150 clones and they were replaced.”
We appreciate these requests for clarification and have rewritten this section to clarify the timeline of the wind event and planting and why lifter was the replacement clone variety.
-Line 101: So only Cherry Wine clones survived after the wind event? If this is correct, I would rephrase this part as: "Only Cherry Wine varieties survived the wind event and the Trophy Wife variety was replaced by Lifter due to its availability. While a fully factorial design was maintained, treatment representation was unbalanced with two hemp varieties: 108 Cherry Wine (C) clones and 42 Lifter (L) clones. "
The revision of this section addressed this concern and clarified what was initially planted, what died in the wind event and why the replacement variety was selected.
-Line 94: I would reword as follow: “The pots were filled to 80% of capacity with a 1:1 ratio mixture of SOAR Potting Mix and Tahoe Propagation Mix (Full Circle Compost), mulched with rice straw and were stored outdoors in the field.”
We have updated the manuscript to include this information: In 2019 the pots were initially filled to 80% of capacity with a 1:1 ratio mixture of SOAR Potting Mix and Tahoe Propagation Mix (Full Circle Compost) and mulched with rice straw for a previous hemp experiment. These same pots, soil and mulch were left in place and used again in 2020.
-Lines 142-150: This part has already been repeated, please check the section and remove the repetitions.
We have rewritten so it is no longer redundant and clarifies the timeline of watering and treatment application.
-Line 164: please remove “(TOF only)
We have retained this parenthetical to differentiate from data-dependent MS/MS mode on our instrument that runs in scan mode until a peak is detected and then collects MS/MS spectra (As opposed to data-independent mode where MS/MS is always running). This detail is important, especially since we do not anticipate all readers of this article to have expertise in mass spectrometry, to clarify (at least once) that the quadrupole was not utilized at all.
-Line 169: I would reword as: “Peaks, analyzed in scan mode, with height greater than 20,000 counts were extracted, and compounds found to have peak height greater than 100,000 in fewer than two individuals were removed.”
-Line 171: I would reword this sentence as: “Calibration curves were generated in scan mode for cannabinoid standards using the same LC-MS conditions described above.”
We appreciate the advice and made these changes as recommended.
-Line 172: the term “quantified” is repeated
Corrected
-Table S3: check the spelling of “cannabinoid”. How did you calculated the parameters of the calibration curve of Δ8-THC in the absence of the commercial standard? I would remove the Calibration Curve Parameters of this compound.
We have fixed the spelling in the text and explained all caveats about calibration curves in the main text (Chemical Analysis section), while maintaining all necessary information in the supplemental tables (as well as making the suggested change below). We used a commercial standard for Δ8-THC, and it is listed as a standard in our methods. It appears there is a misunderstanding here about the absence of Δ8-THCA mentioned below (for which we did not have a standard) with Δ8-THC (for which we did).
- I believe that Table S2 and S3 should be interchanged, since Table S3 is quoted in the text before Table S2.
We have corrected this.
-Line 176, Table S2 and Figure S4: I believe it would be better to reword this sentence as “For THCA compounds, only the Δ9-THCA standard was available; therefore, putative Δ8-THCA concentrations could semi-quantified based on Δ9-THCA calibration curve”
We have made the suggested change.
-Line 176: (3.00x10-7 M – 6.40x10-6 M) what this refers to?
These specify the range of concentrations of the standards prepared for the calibration process – the revised sentence and reference to the table should make this clear.
-Line 192: can you add in the supplementary a representative MS profile with all the compounds considered for the hill numbers test?
Representative EIC chromatograms were provided in the revised supplemental for the compounds considered in high and low diversity samples. Due to large differences in scale, cannabinoids were presented separately from all other compounds.
- Figure 1: the structures of delta9-Tetrahydrocannabinol, delta8-Tetrahydrocannabinol and cannabinol are also incorrect
We have modified the structures to include R groups, which allow for representation of both the acid and neutral structures.
-Figure S2 is cited in the text after Figure S5, please revise the figures number.
We rearranged figure S2 and S5 in the text and in the supplemental.
-Figure S2: please add a heading to the graphs so it is clear whether it is Cherry Wine or Lifter. Also the treatment legend is still unclear, please provide only the three conditions in the legend. The third and fourth graphs are missing the first family name, and some families are still repeated, e.g. Coccinelidae.
We agree that there were several issues with these figures and have made these corrections as recommended.
-Figure S4 and Table S2: I still believe that you should indicate the concentrations as ug/kg instead of ppt and that the statistical significance (e.g. ANOVA) in this case is important, especially in the figure, to evaluate the differences among the treatments.
We appreciate the concern, but this figure provides descriptive statistics and clarity. The concentrations in the manuscript are consistently reported as ppt (parts per thousand, equivalent to μg/mg or 10⁻³), which is the unit suggested by another reviewer and is a practical and widely used unit for the scale of our data and for ecological studies. While μg/kg (equivalent to 10⁻⁹) is another possible unit, it would result in unnecessary clutter that could hinder readability and interpretation. Reporting concentrations as ppt ensures clarity and aligns with conventions for expressing relative concentrations in an ecological context. As for the descriptive statistics, while we agree that ANOVA is a great tool for hypothesis tests focused on comparing means, the hypothesis tests offered by ANOVA were never the goal of this study – the causal models represented by the SEMs include more complete and appropriate hypothesis tests. Furthermore, based on a wealth of literature in statistical journals (e.g., McShane, B.B., Gal, D., Gelman, A., Robert, C. and Tackett, J.L., 2019. Abandon statistical significance. The American Statistician, 73(sup1), pp.235-245…. and many others) we continue to recommend that authors eschew the dichotomy of “statistical significance” and have focused on path coefficients in causal models as appropriate effect sizes.
-Figure S4: Lifter clones were not exposed to wind stress, how is it possible that the hatched bars are also displayed for Lifted clones? Also replace the word hashed with hatched.
The original clones were both Lifter and Cherry wine so there were clones of each variety exposed to wind stress. We have rewritten the experimental design section to clarify this.
-Based on you reply “Table S2 and Figure S4 are reporting different data. Table S2 does not include wind effects, and as such the high standard deviation is reflecting large differences in CBDV between wind stressed and unstressed plants. Also, table S2 separates data into each varietal whereas Figure S4 includes all varietals within each treatment group. Figure S4 error bars report standard error and not standard deviation, so the higher n resulting from combining all varietals together leads to these standard error bars looking much smaller than the standard deviation (s.e. = s.d./sqrt(n)).” I believe that you should specify these details in the captions.
We appreciate this comment that helped us catch the confusion with the labeling of tables (S2 and S3 were switched), which we have corrected. All figure captions clearly specify the estimates of dispersion (i.e. whether they are S.E.M. or S.D.).
-Line 275: you can compare these results with the quantitative data (Figure S4) and the information about the arthropod (Figure S2) obtained in this work. CBD is higher in Cherry Wife variety, does this variety host a higher arthropod diversity? Furthermore, wind stress in Figure S4 does not seem to have a negative impact on CBD content, while Figure 4 seems to suggest this trend. How can this aspect be explained?
These are excellent points. We have clarified in the revision (see Figures 3 and 4 as well as the comment below) that the Cherry wine has higher overall CBD and that these changes in CBD caused lower arthropod diversity. Because causal models reveal both direct and indirect effects, the patterns in raw, descriptive statistics can seem counterintuitive (see references by Judea Pearl), which is why we prefer to only include the hypothesis tests associated with our causal models. We have further explained direct and indirect effects in our figure captions. Beyond the hypothesis tests in the SEM, we are not willing to pile up additional hypothesis tests. Summary statistics in the supplemental figures do not take into account how parameter estimates can change considerably when appropriately modeling them in a causal framework.
-Line 302: I do not think it's true that Lifter variety has a higher CBD content than Cherry Wine. Also, Cherry Wine is characterized by a lower phytochemical diversity as stated in the results section.
We appreciate the reviewer catching this major error and have changed the text to read, “the Lifter variety had lower CBD than the Cherry wine variety, Lifter plants were also characterized by higher phytochemical diversity. This shift is driven by higher cannabinoid concentrations in Cherry wine varieties but lower overall LC-MS peak diversity…”
Reviewer 2 Report
Comments and Suggestions for Authors
The authors made a careful major revision on this manuscript and added some detailed figures to support the results. I am satisfied with these corrections. It is suggested to be accepted after the following minor correction.
Page 7, line 172: “we quantified” should be deleted.
Figure 5: ∆9-THC or ∆9-THCA? Please check.
Page 18, line 336: . is changed to ,
Page 19: Please unify the style of reference. For example, pp is present before the page in some references while absent in other references. There is no page in reference 16. For references 30-32, there should be only one capitalized first letter in the article title.
Author Response
We appreciate the attention to details from all the reviewers and have improved the manuscript again based on their suggestions.
Page 7, line 172: “we quantified” should be deleted.
Corrected.
Figure 5: ∆9-THC or ∆9-THCA? Please check.
We have modified the structures to include R groups, which allow for representation of both the acid and neutral structures.
Page 18, line 336: . is changed to ,
Corrected.
Page 19: Please unify the style of reference. For example, pp is present before the page in some references while absent in other references. There is no page in reference 16. For references 30-32, there should be only one capitalized first letter in the article title.
We have corrected the inconsistencies in the references and appreciate this attention to detail by the reviewer.
Reviewer 3 Report
Comments and Suggestions for Authors
This seems fairly better and reasonable than previous draft
Author Response
We appreciate the attention to details from all the reviewers and have improved the manuscript again based on their suggestions.
Reviewer 4 Report
Comments and Suggestions for Authors
No further comments.
Author Response

(The authors gave the same response as above.)
